# Application of Event Cameras and Neuromorphic Computing to VSLAM: A Survey

**DOI:** 10.3390/biomimetics9070444

**Published:** 2024-07-20

**Authors:** Sangay Tenzin, Alexander Rassau, Douglas Chai

**Affiliations:** School of Engineering, Edith Cowan University, Perth, WA 6027, Australia; s.tenzin@ecu.edu.au (S.T.); d.chai@ecu.edu.au (D.C.)

**Keywords:** Simultaneous Localization and Mapping (SLAM), event camera, neuromorphic processing, robotics, autonomous systems, sensor fusion, real-time processing, machine vision

## Abstract

Simultaneous Localization and Mapping (SLAM) is a crucial function for most autonomous systems, allowing them to both navigate through and create maps of unfamiliar surroundings. Traditional Visual SLAM, also commonly known as VSLAM, relies on frame-based cameras and structured processing pipelines, which face challenges in dynamic or low-light environments. However, recent advancements in event camera technology and neuromorphic processing offer promising opportunities to overcome these limitations. Event cameras inspired by biological vision systems capture the scenes asynchronously, consuming minimal power but with higher temporal resolution. Neuromorphic processors, which are designed to mimic the parallel processing capabilities of the human brain, offer efficient computation for real-time data processing of event-based data streams. This paper provides a comprehensive overview of recent research efforts in integrating event cameras and neuromorphic processors into VSLAM systems. It discusses the principles behind event cameras and neuromorphic processors, highlighting their advantages over traditional sensing and processing methods. Furthermore, an in-depth survey was conducted on state-of-the-art approaches in event-based SLAM, including feature extraction, motion estimation, and map reconstruction techniques. Additionally, the integration of event cameras with neuromorphic processors, focusing on their synergistic benefits in terms of energy efficiency, robustness, and real-time performance, was explored. The paper also discusses the challenges and open research questions in this emerging field, such as sensor calibration, data fusion, and algorithmic development. Finally, the potential applications and future directions for event-based SLAM systems are outlined, ranging from robotics and autonomous vehicles to augmented reality.

## 1. Introduction

A wide and growing variety of robots is increasingly being employed in different indoor and outdoor applications. To support this, autonomous navigation systems have become essential for carrying out many of the required duties [1]. However, such systems must be capable of completing assigned tasks successfully and accurately with minimal human intervention. To increase the effectiveness and efficiency of such systems, they should be capable of navigating to a given destination while simultaneously updating their real-time location and developing a map of the surroundings. Towards this, Simultaneous Localization and Mapping (SLAM) is currently one of the most employed methods for localization and navigation of mobile robots [2]. The concept of SLAM originated from the robotics and computer vision field. SLAM is a joint problem of simultaneously locating the position of the robots while developing a map of their surroundings [3]. It has become a critical technology for tackling the difficulties of allowing machines (autonomous systems) to independently navigate and map unfamiliar surroundings [3,4]. With SLAM, the location and map information of the autonomous systems will be continuously updated in real time. This process can help users in getting the status of the system, as well as serve as a reference in making autonomous navigation-related decisions [3,5]. It helps robots gain autonomy and reduce the requirement for human operation or intervention [3,4]. Moreover, with effective SLAM methods, mobile robots such as vacuum cleaners, autonomous vehicles, aerial drones, and others [2,4] can effectively navigate a dynamic environment autonomously.

The sensor choice affects the performance and efficacy of the SLAM solution [3] and should be decided based on the sensor’s information-gathering capability, power cost, and precision. The primary sensor types commonly utilized in SLAM applications are laser sensors (such as Light Detection and Ranging (LiDAR) sensors) and vision sensors. Laser-based SLAM typically offers higher precision; however, these systems tend to be more expensive and power-hungry [6]. Moreover, they lack semantic information and face challenges in loop closure detection. In environments with a lack of scene diversity, such as uniform corridors or consistently structured tunnels, degradation issues may arise, particularly affecting laser SLAM performance compared to Visual SLAM (VSLAM) [6]. Conversely, VSLAM boasts advantages in terms of cost-effectiveness, compact size, minimal power consumption, and the ability to perceive rich information, rendering it more suitable for indoor settings [6].

In recent decades, VSLAM has gained significant development attention as research has demonstrated that detailed scene information can be gathered from visual data [3,7], as well as due to the increased availability of low-cost cameras [7,8]. In VSLAM, cameras such as monocular, stereo, or RGB-D are used to gather visual information that can be used to solve the localization and map-building problems. These cameras record a continuous video stream by capturing frames of the surrounding environment at a specific rate. The different types of VSLAM systems that have been developed based on these different camera types are detailed in Section Limitations of Frame-Based Cameras in VSLAM, but generally, the classical VSLAM framework follows the steps as shown in Figure 1: sensor data acquisition, visual odometry (VO; also known as front-end), backend filtering/optimization, loop closure, and reconstruction [9]. Sensor data acquisition involves the acquisition and pre-processing of data captured by the sensors (a camera in the case of VSLAM). VO is used to measure the movement of the camera between the adjacent frames (ego-motion) and generate a rough map of the surroundings. The backend optimizes the camera pose received from VO and the result of loop closure in order to generate an efficient trajectory and map for the system. Loop closure determines if the system has previously visited the location to minimize the accumulated drift and update the backend for further optimization. With reconstruction, a map of the system can be developed based on the camera trajectory estimation.

Conventional VSLAM systems gather image data at fixed frame rates, which results in repetitive and often redundant information leading to high computational requirements and other drawbacks [10,11]. Further, they often fail to achieve the expected performance in challenging environments, such as those with high dynamic ranges or light-changing conditions [10,12,13,14,15,16,17], due to constraints such as susceptibility to motion blur, high power consumption, and low dynamic range. These limitations of frame-based cameras will be discussed in more detail in Section Limitations of Frame-Based Cameras in VSLAM, below, but given these issues, research in emerging technologies of event cameras has evolved to attempt to address them. The advent of novel concepts and the production of bio-inspired visual sensors and processors through developments in neuroscience and neuromorphic technologies have brought a radical change in the processes of artificial visual systems [18,19,20]. An event camera (also known as a Dynamic Vision Sensor (DVS) or neuromorphic camera) operates very differently from conventional frame-based cameras; it only generates an output (in the form of timestamped events or spikes) when there are changes in the brightness of a scene [13,18,19]. Figure 2 depicts the three-layer model of a human retina and corresponding event camera pixel circuitry. Compared to regular cameras, event cameras have greater dynamic range, reduced latency, higher temporal resolution, and significantly lower power consumption and bandwidth usage [3,13,14,20,21,22,23,24,25]. However, sensors based on these principles are relatively new to the market and their integration poses some challenges as new algorithms are needed because existing approaches are not directly applicable.

Similarly, in an attempt to further reduce the power cost, the research trends of mimicking the biological intelligence of the human brain and its behavior, known as neuromorphic computing [12,27], are gaining more research focus for application in autonomous systems and robots as an extension to the use of event-based cameras for SLAM [27]. In neuromorphic computing, computational systems are designed by mimicking the composition and operation of the human brain. The objective is to create algorithms and hardware replicating the brain’s energy efficiency and parallel processing capabilities [28]. Unlike von Neumann computers, neuromorphic computers (also known as non-von Neumann computers) consist of neurons and synapses rather than a separate central processing unit (CPU) and memory units [29]. Moreover, as they are fully event-driven and highly parallel, in contrast to traditional computing systems, they can natively deal with spike-based outputs rather than binary data [29]. Furthermore, the advent of neuromorphic processors with various sets of signals to mimic the behavior of biological neurons and synapses [12,30,31] has paved a new direction in the neuroscience field. This enables the hardware to asynchronously communicate between its components and the memory in an efficient manner, which results in less consumption of power in addition to other advantages [12,29,31]. As the computation is based on neural networks, it has become a primarily relevant platform for use in artificial intelligence and machine learning applications to enhance robustness and performance [29,32].

The combination of event cameras and neuromorphic processing, which takes inspiration from the efficiency of the human brain, has the potential to offer a revolutionary approach to improve VSLAM capabilities [18]. The use of event cameras in SLAM systems enables them to handle dynamic situations and fast motion without being affected by motion blur or illumination variations. Event cameras provide high dynamic range imagery and low latency through asynchronous pixel-level brightness change capture [13,18,19,22]. Additionally, neuromorphic processors emulate the brain’s structure and functionality [28], enabling efficient and highly parallel processing, which is particularly advantageous for real-time SLAM operations on embedded devices. This integration would facilitate improved perception, adaptability, and efficiency in SLAM applications, overcoming the limitations of conventional approaches and paving the way for more robust and versatile robotic systems [3]. The successful implementation of these trending technologies is expected to make smart and creative systems capable of making logical analyses at the edge, further enhancing the productivity of the processes, improving precision and minimizing the exposure of humans to hazards [12,27,33]. However, to the best of the authors’ knowledge, there are no existing reviews on the integration of these emerging technologies and there remains a lack of comprehensive reviews encompassing both event cameras and neuromorphic computing in SLAM research. The reviews by [15,18] primarily discussed event cameras and gave only a brief introduction to both SLAM and the application of neuromorphic computing. Similarly, refs. [12,31,34,35] covered neuromorphic computing technology and its challenges; however, no clear direction towards its integration into event-based SLAM was provided. Other review papers [24,36,37,38,39,40] have mentioned the methods and models to be employed in SLAM but did not discuss the combined approach.

The focus and intent of this review article, therefore, is to present a comprehensive exploration of VSLAM techniques, focusing particularly on the limitations of standard frame-based cameras within this application area, the emergence of event cameras as an alternative, and the integration of neuromorphic computing within an event-driven system for enhanced performance. For this review, relevant articles from research databases (such as Web of Science, IEEE Explorer, and Google Scholar) and patents were identified using the keywords SLAM, VSLAM, event camera, and neuromorphic computing, and combinations of those keywords (event-based SLAM, neuromorphic SLAM, and others), to select a set of relevant papers. Additionally, to streamline the search queries, criteria on year range (for example, from 2019) were used to select the most recent and up-to-date articles for the review. Through the thorough analysis and review of articles gathered using these methods, this paper aims to provide the following contributions to the knowledge base in the field of VSLAM technology:A critical analysis and discussion on the methods and technologies employed by traditional VSLAM systems.An in-depth discussion on the challenges and further directions to improve or resolve the identified problems or limitations of traditional VSLAMs.A rationale for and analysis of the use of event cameras in VSLAM to overcome the issues faced by conventional VSLAM approaches.A detailed exploration of the feasibility of integrating neuromorphic processing into event-based VSLAM systems to further enhance performance and power efficiency.

The overall structure of the paper is organized as follows:Introduction: Provides an overview of the problem domain and highlights the need for advancements in VSLAM technology.Frame-based cameras in VSLAM (Section 2): Discusses the traditional approach to VSLAM using frame-based cameras, outlining their limitations and challenges.Event cameras (Section 3): Introduces event cameras and their operational principles, along with discussing their potential benefits and applications across various domains.Neuromorphic computing in VSLAM (Section 4): Explores the application of neuromorphic computing to the VSLAM problem, emphasizing its capability to address performance and power consumption issues commonly encountered within the autonomous systems context.Summary and future directions (Section 5): Provides a synthesis of the key findings from the previous sections and outlines potential future directions for VSLAM research, particularly focusing on the integration of event cameras and neuromorphic processors.

It is hoped that this structure will guide readers through a logical progression from understanding the limitations of traditional camera-based VSLAM approaches to envisioning the potential of cutting-edge technologies such as event cameras and neuromorphic computing in advancing VSLAM capabilities.

## 2. Camera-Based SLAM (VSLAM)

For SLAM implementations, VSLAM is more popular than LiDAR-based SLAM for smaller-scale autonomous systems, particularly unmanned aerial vehicles (UAVs), as it is compact, cost-effective, and less power-intensive [3,7,9,24,37,41]. Unlike the laser-based systems, VSLAM employs various cameras such as monocular, stereo, and RGB-D cameras for capturing the surrounding scene and is being explored by researchers for implementation in autonomous systems and other applications [3,9,24,37,41]. It has gained popularity in the last decade as it has succeeded in retrieving detailed information (color, texture, and appearance) using low-cost cameras, and some progress towards practical implementation in real environments has been made [3,7,18,24]. One prevalent issue encountered in VSLAM systems is the issue of cumulative drift [6]. Minor inaccuracies are produced with every calculation and optimization made by the front end of the SLAM system. These small errors accumulate over the extended durations of uninterrupted camera movement, which eventually causes the estimated trajectory to deviate from the real motion trajectory.

These traditional camera-based VSLAM systems have generally failed to achieve the expected performance in challenging environments such as those with high dynamic ranges or changing lighting conditions [10,12,13,14,15,16,18] due to constraints such as susceptibility to motion blur, noise susceptibility, and low dynamic range, among others. Moreover, the information gathered with traditional VSLAM is inadequate to fulfil the tasks of autonomous navigation and obstacle avoidance, as well as the interaction needs of intelligent autonomous systems in dealing with the human environment [10].

In line with the growing popularity of VSLAM in the last decade, researchers have worked on designing improved algorithms towards making practical and robust solutions for SLAM a reality. However, most of the successfully developed algorithms such as MonoSLAM [42], PTAM [43], DTAM [44], and SLAM++ [45] have been developed for stationary environments, where it is assumed that the camera is the sole moving item in a static environment. This means they are not suitable for applications where both the scene and the object are dynamic [46], like autonomous vehicles and UAVs.

Table 1 provides a summary of the range of VSLAM algorithms that have been developed for testing and implementation in SLAM systems and the different sensor modalities used by each.

### Limitations of Frame-Based Cameras in VSLAM

While there have been some significant successes in the development of VSLAM algorithms for traditional frame-based cameras, some limitations still exist, as described below.

Ambiguity in feature matching: In feature-based SLAM, feature matching is considered a critical step. However, frame-based cameras face difficulty in capturing scenes with ambiguous features (e.g., plain walls). Moreover, data without depth information (as obtained from standard monocular cameras) makes it even harder for the feature-matching process to distinguish between similar features, which can lead to potential errors in data association.Sensitivity to lighting conditions: The sensitivity of traditional cameras to changes in lighting conditions affects the features and makes it more challenging to match features across frames consistently [7]. This can result in errors during the localization and mapping process.Limited field of view: The use of frame-based cameras can be limited due to their inherently limited field of view. This limitation becomes more apparent in environments with complex structures or large open spaces. In such cases, having multiple cameras or additional sensor modalities may become necessary to achieve comprehensive scene coverage, but this can lead to greatly increased computational costs as well as other complexities.Challenge in handling dynamic environments: Frame-based cameras face difficulties when it comes to capturing dynamic environments, especially where there is movement of objects or people. It can be challenging to track features consistently in the presence of moving entities, and other sensor types such as depth sensors or inertial measurement units (IMUs) must be integrated, or additional strategies must be implemented to mitigate those challenges. Additionally, in situations where objects in a scene are moving rapidly, particularly if the camera itself is on a fast-moving platform (e.g., a drone), then motion blur can significantly degrade the quality of captured frames unless highly specialized cameras are used.High computational requirements: Although frame-based cameras are typically less computationally demanding than depth sensors such as LiDAR, feature extraction and matching processes can still necessitate considerable computational resources, particularly for real-time applications.

## 3. Event Camera-Based SLAM

Due to the limitations of traditional cameras highlighted in the previous section, event cameras have begun to be explored by researchers in the field of SLAM. Event cameras have gained attention due to their unique properties, such as high temporal resolution, low latency, and high dynamic range. However, tackling the SLAM issue using event cameras has proven challenging due to the inapplicability of traditional frame-based camera methods and concepts such as feature detection, matching, and iterative image alignment. Events are fundamentally distinct from images as illustrated in Figure 3, which shows the differing output of frame-based cameras relative to event cameras. This necessitates the development of novel techniques for solution of the SLAM problem. The primary task has been to devise approaches that harness the unique advantages of event cameras, demonstrating their efficacy in addressing challenging scenarios that are problematic for current frame-based cameras. A primary aim when designing the methods has been to preserve the low latency nature of event data, thereby estimating a state for every new event. However, individual events lack sufficient data to create a complete state estimate, such as determining the precise position of a calibrated camera with six degrees of freedom (DoF). Consequently, the objective has shifted to enabling each event to independently update the system’s state asynchronously [18].

Utilizing event cameras’ asynchrony and high temporal resolution, SLAM algorithms can benefit from reduced motion blur and improved visual perception in dynamic environments, ultimately leading to more robust and accurate mapping and localization results [61]. It can enhance the reconstruction of 3D scenes and enable tracking of fast motion with high precision. Furthermore, a low data rate and reduced power consumption compared to traditional cameras make them ideal for resource-constrained devices in applications such as autonomous vehicles, robotics, and augmented reality [61]. Moreover, it can be used to significantly increase the frame rate of low-framerate video while occupying significantly less memory space than conventional camera frames, enabling efficient and superior-quality video frame interpolation [22].

The integration of event cameras in SLAM systems opens new possibilities for efficient and accurate mapping, localization, and perception in dynamic environments, while also reducing power consumption and memory usage. These enhanced capabilities also enable new opportunities and applications, some of which are discussed in more detail in Section 3.4.

### 3.1. Event Camera Operating Principles

Standard cameras and event cameras have significant differences when it comes to their working principle and operation [18,21]. Conventional cameras record a sequence of images at a predetermined frames per second (fps) rate, capturing intensity values for every pixel in every frame. On the other hand, event cameras record continuous-time event data, timestamped with microsecond resolution, with an event representing a detected change in pixel brightness [18,20,61,62]. Each pixel continuously updates the log intensity, and this is monitored for any notable changes in its value. If the value changes (either high or low) more than a certain threshold, an event will be generated [18]. The process of generating events by the event camera with the change in illumination is shown in Figure 4.

An event is represented as a tuple, $e_{k}=(x_{k},\;y_{k},\;t_{k},\;p_{k})$, where ($x_{k},y_{k}$) denotes the pixel coordinates that caused the event, $t_{k}$ is the timestamp, and $p_{k}=\pm 1$ denotes the polarity or direction of the change in brightness [61]. Events are transferred from the pixel array to the peripheral and back again over a shared digital output bus, typically using the address-event representation (AER) readout technique [63]. Saturation of this bus, however, can occur and cause hiccups in the event transmission schedule. Event cameras’ readout rates range from 2 MHz to 1200 MHz, depending on the chip and hardware interface type being used [18].

Event cameras are essentially sensors that react to motion and brightness variations in a scene. They produce more events per second when there is greater motion. The reason is that every pixel modifies the rate at which it samples data using a delta modulator in response to variations in the log intensity signal it tracks. These sensors can respond to visual stimuli rapidly because of the sub-millisecond latency and microsecond precision timestamped events. Surface reflectance and scene lighting both affect how much light a pixel receives. A change in log intensity denotes a proportional change in reflectance in situations where illumination is largely constant. The primary cause of these reflectance changes is the motion of objects in the field of view. Consequently, the brightness change events captured inherently possess an invariance to changes in scene illumination.

#### 3.1.1. Event Generation Model

At each pixel position $u_{k}$ the event camera sensor first records and stores the logarithmic intensity of brightness, or $L(u_{k}\;)=log(I(u_{k}))$ and then continuously monitors this intensity value. The camera sensor at the pixel position $u_{k}=(x_{k},\;y_{k})$ generates an event, denoted by $e_{k},$ at time $t_{k}$ when the difference in intensity, $∆L(u_{k},\;t_{k})$ exceeds a threshold, *C*, which is referred to as contrast sensitivity.
(1)$$∆L\left(u_{k},\;t_{k}\right)=L\left(u_{k},\;t_{k}\right)-L(u_{k},\;t_{k}-∆t_{k})=p_{k}C$$

The last timestamp that was recorded in this context is $t_{k}-∆t_{k}$, which occurs when an event is triggered at the pixel $u_{k}$. The camera sensor then creates new events by iterating through the procedure to detect any changes in brightness at this pixel, updating the stored intensity value $L\left(u_{k},\;t_{k}\right).$ The adjustable parameter *C*, or temporal contrast sensitivity, is essential to the camera’s functioning; it is usually set in the range of 10% to 15%. A high contrast sensitivity can result in fewer events produced by the camera and potential information loss, whereas a low contrast sensitivity may cause an excessive number of noisy events.

#### 3.1.2. Event Representation

The event camera records brightness variations at every pixel, producing a constant stream of event data. The low information content in each record and the sparse temporal nature of the data make the processing difficult. Filter-based methods directly process raw event data by combining it with sequential data, but they come with a high computational cost because they must update the camera motion for every new event. To mitigate this problem, alternative methods employ representations that combine event sequences and approximate camera motion for a collection of occurrences, achieving an equilibrium between computing cost and latency.

Common event representations for event-based VSLAM algorithms are described below:

Individual event: On an event-by-event basis, each event $e_{k}=\left(x_{k},\;y_{k},\;t_{k},\;p_{k}\right)$ may be directly utilized in filter-based models, such as probabilistic filters [64] and spiking neural networks (SNNs) [65]. With every incoming event, these models asynchronously change their internal states, either by recycling states from earlier events or by obtaining new information from outside sources, such as inertial data [18]. Although filter-based techniques can produce very little delay, they generally require a significant amount of processing power.

Packet: The event packet, also known as the point set, is an alternate representation used in event cameras. It stores an event data sequence directly in a temporal window of size N and is stated as follows:(2)$$\varepsilon ={\left\{e_{k}\right\}}_{k}^{N}=1$$

Event packets maintain specific details like polarity and timestamps, just like individual events do. Event packets facilitate batch operations in filter-based approaches [66] and streamline the search for the best answers in optimization methods [67,68] because they aggregate event data inside temporal frames. There are several variations of event packets, including event queues [67] and local point sets [68].

Event frame: A condensed 2D representation of an event that gathers data at a single pixel point is called an event frame. Assuming consistent pixel coordinates, this representation is achieved by transforming a series of events into an image-like format that is used as input for conventional frame-based SLAM algorithms [18].

Time surface: The time surface (TS), also called the surface of active events (SAE), is a 2D representation in which every pixel contains a single time value, often the most recent timestamp of the event that occurred at that pixel [18]. A spatially structured visualization of the temporal data related to occurrences throughout the camera’s sensor array is offered by the time surface. Due to its ability to trace the time of events at different points on the image sensor, this representation can be helpful in a variety of applications, such as visual perception and motion analysis [18].

Motion-compensated event frame: A motion-compensated event frame refers to a representation in event cameras where the captured events are aggregated or accumulated while compensating for the motion of the camera or objects in the scene [18]. Unlike traditional event frames that accumulate events at fixed pixel positions, motion-compensated event frames consider the dynamic changes in the scene over time. The events contributing to the frame are not simply accumulated at fixed pixel positions, but rather the accumulation is adjusted based on the perceived motion in the scene. This compensation can be performed using various techniques, such as incorporating information from inertial sensors, estimating camera motion, or using other motion models [18].

Voxel grid: A voxel grid can be used as a representation of 3D spatial information extracted from the events captured by the camera. Instead of traditional 2D pixel-based representations, a voxel grid provides a volumetric representation of the environment [18], allowing for 3D scene reconstruction, mapping, and navigation.

3D point set: Events within a spatiotemporal neighborhood are regarded as points in 3D space, denoted as $\left(x_{k},\;y_{k},\;t_{k}\right)\epsilon R$. Consequently, the temporal dimension is transformed into a geometric one. Plane fitting [69] or Point Net [70] are two point-based geometric processing methods that use this sparsely populated form.

Point sets on the image plane: On the picture plane, events are viewed as a dynamic collection of 2D points. This representation is frequently used in early shape-tracking methods that use methods like mean-shift or iterative closest point (ICP) [71,72,73,74,75], in which the only information needed to follow edge patterns is events.

### 3.2. Method

To process the events, a relevant and valid method is required depending on the event representation and specifics of the hardware platform. Moreover, the relevant information from event data that needs to be extracted to fulfil the required task depends on the application and algorithm being utilized [18]. However, the efficacy of such efforts varies significantly based on the nature of the application and the unique demands it places on the data being extracted [18]. Figure 5 presents an overview of common methods used for event-based SLAM.

#### 3.2.1. Feature-Based Methods

The feature-based VSLAM algorithms comprise two main elements: (1) the extraction and tracking of features, and (2) the tracking and mapping of the camera. During the feature extraction phase, resilient features, immune to diverse factors such as motion, noise, and changes in illumination, are identified. The ensuing feature tracking phase is employed to link features that correspond to identical points in the scene. Leveraging these associated features, algorithms for camera tracking and mapping concurrently estimate the relative poses of the camera and the 3D landmarks of the features.

##### Feature Extraction

Methods for feature extraction identify shape primitives within the event stream, encompassing features based on points and lines. Point-based features denote points of significance, such as the intersection of event edges. Various methods for point-based feature extraction, particularly corners, in the context of event cameras, have been used in the last decade or so. Traditional techniques involve employing algorithms like local plane fitting [76,77], frame-based corner detectors (e.g., eHarris [78], CHEC [79], eFAST [80]), and extensions of the Harris and FAST detectors to different event representations [81,82,83]. These methods, however, suffer from computational complexity, sensitivity to motion changes, and susceptibility to noise in event cameras [61]. To address these challenges, learning-based approaches [84,85] have been proposed, including the use of speed-invariant time surfaces and recurrent neural networks (RNNs) to enhance corner detection stability by implicitly modeling motion-variant patterns and event noise.

On the other hand, line-based features consist of clusters of events situated along straight lines. Several algorithms including classical methods like the Hough transformation and Line Segment Detector (LSD) [86] have been employed. Some approaches leverage spatiotemporal relationships [87] in event data, while others use external IMU [88] data to group events. Examples include a spiking Hough transformation algorithm using spiking neurons [89] and extending the Hough transformation to a 3D point-based map [90] for improved performance. Event-based VO with point and line features (PL-EVIO) leverages line-based event features to add more structure and constraint while efficiently handling the point-based event and picture characteristics; ref. [91] directly applies the LSD algorithm to motion-compensated event streams, while the Event-based Line Segment Detector (ELiSeD) [92] computes event orientation using the Sobel filter. Other methods use optical flow [93] or plane-fitting algorithms [87] to cluster events and extract lines, demonstrating different techniques for line-based feature extraction from event data.

##### Feature Tracking

When working with event-based data, feature-tracking algorithms are utilized to link events to the relevant features. Motion trajectories, locations, and 2D rigid body transformations are examples of parametric models of feature templates that these algorithms update [61]. Methods include parametric transformations like the Euclidean transformation and descriptor matching for feature correspondences. Deep learning approaches use neural networks to predict feature displacements. Euclidean transformations model positions and orientations of event-based features, and tracking involves ICP algorithms [94] with enhancements like Euclidean distance weighting and 2D local histograms to improve accuracy and reduce drift. Some trackers, such as the Feature Tracking using Events and Frames (EKLT) tracker [95], align local patches of the brightness incremental image from event data with feature patterns and estimate brightness changes using the linearized Edge Gradient Method (EGM). Feature tracking often involves modeling feature motions on the image plane, with methods using expectation-maximization (EM) optimization steps [81,82] and the Lucas–Kanade (LK) optical flow tracker [83,91]. Continuous curve representations, like Bezier curves [96] and B-splines [97], are explored to address linear model assumptions. Multi-hypothesis methods [67,98] are proposed to handle event noise by discretizing spatial neighborhoods into hypotheses based on distance and orientation. Various techniques include using feature descriptors for direct correspondence establishment and building graphs with nodes representing event characteristics for tracking based on their discrete positions on the image plane [99,100]. Traditional linear noise models are contrasted with deep learning methods that implicitly model event noise [101].

##### Camera Tracking and Mapping

VSLAM algorithms, particularly those adapted for event-based tracking and mapping, introduce two main paradigms: one where 3D maps are initialized, and tracking and mapping are performed in parallel threads, and another where tracking and mapping are carried out simultaneously through joint optimization. The former offers computational efficiency, while the latter helps prevent drift errors. Event-based VSLAM approaches in camera tracking and mapping are categorized into four types: conventional frame-based methods, filter-based methods, continuous-time camera trajectory methods, and spatiotemporal consistency methods.

Conventional frame-based methods adapt existing VSLAM algorithms for event-based tracking and mapping using 2D image-like event representation. Various techniques, such as reprojection error and depth estimation, are employed for camera pose estimation. Event-based Visual Inertial Odometry (EVIO) [82] methods utilize IMU pre-integration and sliding-window optimization. Filter-based methods handle asynchronous event data using a state defined as the current camera pose and a random diffusion model as the motion model. These methods correct the state using error measurements, with examples incorporating planar features and event occurrence probabilities. Line-based SLAM methods update filter states during camera tracking and use the Hough transformation to extract 3D lines. Continuous-time camera trajectory methods represent the camera trajectory as a continuous curve, addressing the parameterization challenge faced by filter-based methods. Joint optimization methods based on incremental Structure from Motion (SfM) are proposed to update control states and 3D landmarks simultaneously. Spatiotemporal consistency methods introduce a constraint for events under rotational camera motion, optimizing motion parameters through iterative searches and enforcing spatial consistency using the trimmed ICP algorithm.

#### 3.2.2. Direct Method

Direct methods do not require explicit data association, as opposed to feature-based approaches, and instead directly align event data in camera tracking and mapping algorithms. Although frame-based direct approaches use pixel intensities between selected pixels in source and target images to estimate relative camera poses and 3D positions, they are not applicable to event streams because of their asynchronous nature and the absence of brightness change information in the event data. Two kinds of event-based direct techniques—event-image alignment and event representation-based alignment—have been developed to overcome this difficulty. The Edge Gradient Method (EGM) is used by event-image alignment techniques, such as those demonstrated by [64,102], to take advantage of the photometric link between brightness variations from events and absolute brightness in images. Event representation-based alignment techniques [16,103] use spatiotemporal information to align events by transforming event data into 2D image-like representations.

Photometric consistency between supplementary visual images and event data is guaranteed by event-image alignment techniques. To estimate camera positions and depths, these approaches [64,104,105] correlate event data with corresponding pixel brightness levels. Filter-based techniques are employed in direct methods to process incoming event data. For example, one approach [105] uses two filters for camera pose estimation and image gradient calculation under rotational camera motion. The first filter utilizes the current camera pose and Gaussian noise for motion modeling, projecting events to a reconstructed reference image and updating state values based on logarithmic brightness differences. The second filter estimates logarithmic gradients using the linearized Edge Gradient Method (EGM) and employs interleaved Poisson reconstruction for absolute brightness intensity recovery. An alternate method to improve robustness is to estimate additional states for contrast threshold and camera posture history, then filter outliers in event data using a robust sensor model with a normal-uniform mixed distribution [104].

Several techniques [66,106] are proposed for estimating camera posture and velocity from event data. One method [66] considers the fact that events are more frequent in areas with large brightness gradients and maximizes a probability distribution function proportional to the magnitude of camera velocity and image gradients. An alternative method [106] makes use of the linearized EGM to determine the camera motion parameters, calculating both linear and angular velocity by taking the camera’s velocity direction into account. Non-linear optimization is used in some techniques [102,107] to process groupings of events concurrently to reduce the computational cost associated with updating camera positions on an event-by-event basis. These methods estimate camera posture and velocity simultaneously by converting an event stream to a brightness incremental image and aligning it with a reference image. While one approach [107] uses the mapping module’s provided photometric 3D map as an assumption, another [102] uses Photometric Bundle Adjustment (PBA) to fine-tune camera positions and a 3D structure by transferring depth values between keyframes.

To guarantee photometric consistency, event-image alignment techniques rely on extra information such as brightness pictures and a photometric 3D map with intensities and depths. On the other hand, event representation-based alignment techniques map onto the structure of the frame-based direct method, transforming event data into representations that resemble 2D images. A geometric strategy based on edge patterns is presented by the event-based VO (EVO) [103] method for aligning event data. It aligns a series of events with the reference frame created by the 3D map’s reprojection in its camera tracking module by converting them into an edge map. The mapping module rebuilds a local semi-dense 3D map without explicit data associations using Event-Based Multi-View Stereo (EMVS) [108].

To take advantage of the temporal information contained in event data, Event-Based Stereo Visual Odometry (ESVO) [16] presents an event-event alignment technique on a time surface (TS). A TS is interpreted by ESVO as an anisotropic distance field in its camera tracking module, which aligns the support of the semi-dense map with the latest events in the TS. The task of estimating the camera position is expressed as a minimization problem by lining up the support with the negative TS minima. To maximize stereo temporal consistency, ESVO uses a forward-projection technique to reproject reference frame pixels to stereo TS during mapping. By combining the depth distribution in neighborhoods and spreading earlier depth estimates, a depth filter and fusion approach are created to improve the depth estimation. A different approach [22] suggests a selection procedure to help the semi-dense map get rid of unnecessary depth points and cut down on processing overhead. Furthermore, it prevents the degradation of ESVO in scenarios with few generated events by fusing IMU data with the time surface using the IMU pre-integration algorithm [22]. In contrast, Depth-Event Camera Visual Odometry (DEVO) [109] uses a depth sensor to enhance the creation of a precise 3D local map that is less affected by erratic events in the mapping module.

#### 3.2.3. Motion Compensation Methods

Using the event frame as the fundamental event representation, motion-compensation techniques are based on event alignment. To provide clear images and lessen motion blur over a longer temporal window, these algorithms optimize event alignment in the motion-compensated event frame to predict camera motion parameters. On the other hand, there is a chance of unfavorable results, including event collapse, in which a series of events builds up into a line or a point inside the event frame. Contrast Maximization (CMax), Dispersion Minimization (DMin), and Probabilistic Alignment techniques are the three categories into which the approaches are divided.

Using the maximum edge strengths in the Image Warping Error (IWE), the CMax framework [110] aims to align event data caused by the same scene edges. Optimizing the contrast (variance) of the IWE is the next step in the process, which entails warping a series of events into a reference frame using candidate motion parameters. In addition to improving edge strengths, this makes event camera motion estimation easier.

The DMin methods utilize entropy loss on the warped events to minimize the average event dispersion, strengthening edge structures. They do so by warping events into a feature space using the camera motion model. The potential energy and the Sharma–Mittal entropy are used to calculate the entropy loss. The feature vector undergoes a truncated kernel function-based convolution, which leads to a computational complexity that increases linearly with the number of events. Furthermore, an incremental variation of the DMin technique maximizes the measurement function within its spatiotemporal vicinity for every incoming event.

The possibility that event data would correspond to the same scene point is assessed using a probabilistic model that was recently established in [111]. The pixel coordinates of an event stream are rounded to the nearest neighbor using a camera motion model to create a reference timestamp. The Poisson random variable is used to represent the count of warped events at each pixel, while the spatiotemporal Poisson point process (ST-PPP) model is used to represent the probability of all the warped events together. Next, by maximizing the ST-PPP model’s probability, the camera motion parameters are approximated.

#### 3.2.4. Deep Learning Methods

Deep learning techniques have been widely used in computer vision applications in recent years, and they have shown a great deal of promise in VSLAM algorithms [112,113,114,115,116]. However, typical deep neural networks (DNNs) including multilayer perceptron networks (MLPs), convolutional neural networks (CNNs), and recurrent neural networks (RNNs) have difficulties due to the sparse and asynchronous nature of event data collected by event cameras. Currently, available DNNs often require conversion to voxel grids [117] or event-frame-based representations [69] to process event data. Conversely, individual event data can be processed directly and without pre-processing via SNNs. Supervised and unsupervised learning techniques are additional categories of event-based deep learning.

The goal of supervised deep learning techniques is to minimize the discrepancies between the ground truth and the predicted poses and depths. Using a CNN to extract features from the event frame and a stacked spatial Long Short-Term Memory network (LSTM) to merge it with motion history is one method of regressing camera poses from sequences of event data [71]. Nevertheless, this approach has difficulties when it comes to processing collected events and estimating a camera attitude for a subset of event data inside each accumulation of events. Another method for addressing this is a convolutional SNN for preprocessing-free continuous-time camera posture regression [65].

In unsupervised deep learning methods, depth values and ground truth camera postures are not required for training. Rather, they employ supervisory signals, including photometric consistency, which are acquired through the process of back-warping adjacent frames utilizing the depth and pose predictions of DNNs inside the multi-view geometric constraint.

### 3.3. Performance Evaluation of VSLAM Systems

To assess the relative effectiveness of alternative SLAM solutions, reliable evaluation metrics are needed. This section discusses some of the existing metrics and their applicability to event camera-based SLAM implementations.

#### 3.3.1. Event Camera Datasets

The availability of suitable datasets plays a crucial role in testing and validating the performance of novel systems. In this regard, for the evaluation of event camera-based systems, relevant datasets must be prepared from the images or videos captured using an event camera. Neuromorphic vision datasets follow an event-driven processing paradigm represented by binary spikes and have rich spatiotemporal components compared to traditional frame-based datasets [118]. In general, there are two kinds of neuromorphic datasets, DVS-converted (converted from frame-based static image datasets) and DVS-captured datasets [118]. Although DVS-converted (frameless) datasets can contain more temporal information as compared to the original dataset, they come with certain drawbacks (full temporal information cannot be obtained) and are generally considered not to be a good option for benchmarking SNNs [119,120]. Moreover, it has been observed that spike activity decreases in deeper layers of spiking neurons when they are trained on such data, which results in performance degradation during the training [121]. Conversely, DVS-captured datasets generate spike events naturally, which makes it a more suitable sensor input for SNNs [118,121,122].

Several datasets have been developed to facilitate the evaluation of event-based cameras and SLAM systems, as mentioned in [61]. The early datasets, such as the one introduced in [123], offer sequences captured by handheld event cameras in indoor environments, alongside ground truth camera poses obtained from motion capture systems, albeit limited to low-speed camera motions in small-scale indoor settings. Similarly, the RPG dataset [124] also focuses on indoor environments, utilizing handheld stereo event cameras, but is constrained by similar limitations. In contrast, the MVSEC dataset [74] represents a significant advancement, featuring large-scale scenarios captured by a hexacopter and a driving car, encompassing both indoor and outdoor environments with varied lighting conditions. Another notable dataset, the Vicon dataset reported in [91], incorporates event cameras with different resolutions to capture high-dynamic-range scenarios under challenging lighting conditions. Moreover, recent advancements have led to the release of advanced event-based SLAM datasets [102,125,126,127,128] like the UZH-FPV dataset [125], which employs a wide-angle event camera attached to a drone to capture high-speed camera motions in diverse indoor and outdoor environments, and the TUM-VIE dataset [126], which utilizes advanced event cameras to construct stereo visual-inertial datasets spanning various scenarios from small- to large-scale scenes with low-light conditions and high dynamic range.

#### 3.3.2. Event-Based SLAM Metrics

In assessing the performance of SLAM algorithms, particularly in terms of camera pose estimation, two primary metrics are commonly utilized: the absolute trajectory error (ATE) and the relative pose error (RPE) [129]. ATE quantifies the accuracy of camera poses relative to a world reference, measuring translational and rotational errors between estimated and ground truth poses. Conversely, RPE evaluates the consistency of relative camera poses between consecutive frames. ATE offers a comprehensive assessment of long-term performance, while RPE provides insights into local consistency. Notably, some studies adjust positional error measurements concerning mean scene depth or total traversed distance for scale invariance [91,104]. Additionally, alternative metrics [117] like average relative pose error (ARPE), average relative rotation error (ARRE), and average endpoint error (AEE) are suggested for evaluating translational and rotational differences. ARPE measures the geodesic distance between two rotational matrices, whereas AEE and ARPE quantify the position and orientation differences between two translational vectors, respectively. Average linear and angular velocity errors can also serve as alternative metrics for pose estimation. For depth estimation, the average depth error at various cut-offs up to fixed depth values is commonly employed, allowing for comparisons across diverse scales of 3D maps.

#### 3.3.3. Performance Comparison of SLAM Methods

To evaluate the state-of-the-art methods of SLAM, depth and camera pose estimation quality are additional metrics that can be used to make a performance comparison. In the following section, qualitative analyses based on the existing literature were presented.

##### Depth Estimation

In the study reported in [61], three DNN-based monocular depth estimation techniques have been assessed and compared to the most advanced conventional approaches, which are MegaDepth [75,117], E2Depth [72], and RAM [73]. These techniques were trained using the MVSEC dataset’s outdoor_day *2* sequence [74], and the average depth errors at various maximum cutoff depths (such as 10 m, 20 m, and 30 m) were compared.

According to the results of [61], event-based approaches perform better than frame-based methods when handling fast motion and poor light. MegaDepth’s accuracy decreased in nighttime outdoor_night sequences taken from moving vehicles because of motion blur and a constrained dynamic range. However, it was discovered that using the reconstructed images made from event streams improved the performance. On average, depth mistakes are regularly 1–2 m lower with an unsupervised approach [117] than with MegaDepth. Ref. [61] mentioned that the addition of ground truth labels and more training on artificial datasets were found to increase E2Depth’s efficacy. Further improvements over these event-based techniques are shown by RAM, which combines synchronous intensity images with asynchronous event data. This implies that using static features that are taken from intensity images can improve the efficiency of event-based methods.

##### Camera Pose Estimation

Rotating sequences [123] can be used to evaluate motion compensation algorithms by measuring the root mean square (RMS) of the angular velocity errors. With the least amount of temporal complexity among the assessed techniques, CMax [110] was discovered to exhibit good performance for the 3-DoF rotational motion of event cameras. With the addition of entropy minimization on projected events, DMin [130] improves CMax’s performance in high-dimensional feature spaces by about 20%. However, DMin comes at a significant computational expense. This problem was addressed by Approximate DMin [130], which uses a shortened kernel for increased efficiency. With a 39% improvement in the shape sequence, an alternate method using a probabilistic model, ST-PPP [111], achieved the best performance of all the methods studied.

To assess the performance of both motion-compensation and deep learning techniques on the outdoor day 1 sequence in [117], metrics such as ARPE, ARRE, and AEE were used. It was discovered that DMin [131] performs best when the dispersion of back-projected events in 3D space is kept to a minimum. Additionally, Approximate DMin has reduced the time complexity and outperformed the standard DMin by about 20%. However, the online version of DMin has produced inferior results because of its event-by-event processing. It was discovered that deep learning techniques outperformed motion-compensation techniques [69].

Research has employed boxes [123] and pipe [104] sequences to measure positional mistakes with mean scene depth and orientation errors, to compare the two event-image alignment techniques. Utilizing a filter-based approach that takes advantage of the photometric link between brightness change and absolute brightness, ref. [104] demonstrated very good results. On the other hand, ref. [107] aligned two brightness incremental photos using least-squares optimization to produce even better results.

The RPG dataset [16] has been used to evaluate several EVO algorithms with respect to positional and orientation errors. EVO [103] performed well in a variety of sequences, but it had trouble keeping up with abrupt changes in edge patterns. Outperforming EVO, Ultimate SLAM (USLAM) [83] improved feature-based VO by fusing pictures and inertial data with event data. When it comes to camera pose estimation, ESVO [16] outperformed USLAM and provided more accurate depth estimation from stereo event cameras; however, it still lagged behind frame-based algorithms like Direct Sparse Odometry (DSO) [132] and ORB-SLAM2 [58]. By using photometric bundle correction, Event-aided DSO (EDSO) [102] attained performance that is equivalent to DSO. Additionally, when the reconstructed images from E2VID [133] are taken as an input, DSO achieved better performance in the rpg_desk sequence. Nevertheless, DSO has trouble with high-texture sequences because of E2VID reconstruction problems.

Additionally, the assessment of many EVIO techniques was conducted using the VICON dataset [91], emphasizing positional inaccuracies in relation to the ground truth trajectory’s overall trajectory length. When it comes to combining event data with IMU data and intensity images, USLAM underperformed the frame-based VIO algorithms (SOTA) [85]. With event-corner feature extraction, tracking methods, and sliding-windows graph-based optimization, EIO [134] improved performance. Additionally, PL-EVIO [91] outperformed both event-based and frame-based VIO techniques by extending line-based features in event data and point-based features in intensity images.

### 3.4. Applications of Event Camera-Based SLAM Systems

Due to their unique advantages, event cameras are gaining increasing attention in various fields, including robotics and computer vision. The utilization of event cameras in the SLAM field has the potential to enable several valuable applications in a variety of fields, as discussed below.

#### 3.4.1. Robotics

Event-based SLAM systems have the transformative potential to empower robots with autonomous navigation capabilities even in the most challenging and cluttered environments. By leveraging the asynchronous and high-temporal-resolution data provided by event-based cameras, these systems can offer robots a nuanced understanding of their surroundings, enabling them to navigate with significantly improved precision and efficiency [3,18,61]. Unlike traditional SLAM methods [3,39], event-based SLAMs excel in capturing rapid changes in the environment, allowing robots to adapt swiftly to dynamic obstacles and unpredictable scenarios. This heightened awareness not only enhances the safety and reliability of robotic navigation [18,135], but also opens doors to previously inaccessible environments where real-time responsiveness is paramount.

Obstacle avoidance represents a critical capability in the realm of robotic navigation [7,135] and event-based cameras offer potential advantages for the real-time perception of dynamic obstacles [18]. Event-based sensors will enable robots to swiftly detect and respond to changes in their environment, facilitating safe traversal through complex and challenging landscapes. By continuously monitoring their surroundings with a high temporal resolution [3,22], event-based cameras can enable robots to navigate complex dynamic environments, avoiding collisions and hazards in real time. This capability would not only enhance the safety of robotic operations in dynamic environments, but also unlock new possibilities for autonomous systems to be integrated into human-centric spaces, such as high-traffic streets or crowded indoor environments.

Event-based SLAM systems also provide advantages for tracking moving objects in various critical applications [3,22]. The ability to monitor and follow dynamic entities is important in many applications including navigation in dynamic environments or object manipulation tasks. Event-based cameras, due to their rapid response times and precise detection capabilities, can theoretically be used to capture the motion of objects accurately and efficiently [18]. This real-time tracking functionality will not only enhance situational awareness capability, but also facilitate timely autonomous decision-making processes in dynamic and time-sensitive scenarios.

#### 3.4.2. Autonomous Vehicles

The integration of event-based SLAM systems can provide benefits in the realm of self-driving cars [136]. The unique characteristics of event-based cameras with regards to high temporal resolution and adaptability to dynamic lighting conditions, in conjunction with other sensors, could provide autonomous vehicles [136,137] with improved capability to navigate through challenging scenarios such as in low light or during adverse weather conditions.

Effective collision avoidance systems are vital for the safe operation of autonomous vehicle technology [138], and the integration of event-based cameras has the potential to enhance these systems. By leveraging the unique capabilities of event-based cameras, autonomous vehicles can achieve real-time detection and tracking of moving objects with high levels of precision and responsiveness. By providing high-temporal-resolution data, event-based cameras offer a granular understanding of dynamic traffic scenarios, potentially improving the ability of vehicles to avoid hazardous situations.

#### 3.4.3. Virtual Reality (VR) and Augmented Reality (AR)

With their high temporal resolution and low latency, event camera-based SLAM systems could provide advantages for the accurate inside-out real-time tracking of head movements or hand gestures [18,139], which are important capabilities for immersive VR systems. Their low power requirements would also provide significant benefits for wireless headsets.

Event-based SLAM systems could also provide advantages in the realm of spatial mapping, particularly for augmented reality (AR) applications [7]. Their ability to capture changes in the environment with high temporal resolution, and with robustness to variations in lighting, should enable event-based cameras to create accurate spatial maps in a variety of conditions.

## 4. Application of Neuromorphic Computing to SLAM

The previous section identified the feasibility and potential benefits that can be realized through the application of event cameras to the VSLAM problem. The true potential of event cameras is not readily realized with traditional computing systems, however, because the processing of the event data is computationally expensive and usually requires additional hardware such as GPUs. A more promising pathway exists through the application of neuromorphic computing approaches. The input and output of event cameras are natively compatible with neuromorphic systems, and this integration has the potential to bring about radical change.

Machine learning algorithms have become increasingly powerful and have shown great success in various scientific and industrial applications due to the development of increasingly powerful computers and smart systems. Influenced by the hierarchical nature of the human visual system, deep learning techniques have undergone remarkable advancement [140]. Even with these developments, however, the mainstream machine learning (ML) models in robotics can still not perform tasks with human-like ability, especially in tasks requiring fine motor control, quick reflexes, and flexibility in response to changing environments. There are also significant scalability and deployment issues with these standard machine learning models due to their computational complexity. It is becoming clear that a different paradigm is needed.

The difference in power consumption between the human brain and current AI technology is striking when one realizes that a clock-based computer operating a “human-scale” brain simulation in theory would need about 12 gigawatts, but the human brain only uses 20 Watts [141]. The artificial discretization of time imposed by mainstream processing and sensor architectures [142], which depend on arbitrary internal clocks, is a major barrier to the upscaling of intelligent interactive agents. To process the constant inputs from the outside world, clock frequencies must be raised. However, with present hardware, obtaining such high frequencies is not efficient and practicable for large-scale applications. Biological entities use spikes for information processing to digest information at a high rate of efficiency, which improves their perception and interaction with the outside world. In the quest for computer intelligence that is comparable to that of humans, one difficulty is to replicate the effective neuro-synaptic architecture of the physical brain. Several technologies and techniques aimed at more accurately mimicking the biological behavior of the human brain have been developed because of the considerable exploration of this area in recent years. This conduct is marked by quick response times and low energy use. Neuromorphic computing, sometimes referred to as brain-inspired computing, is one notable strategy in this quest.

A multidisciplinary research paradigm called “neuromorphic computing” investigates large-scale processing devices that use spike-driven communication to mimic natural neural computations. When compared to traditional methods, it has several advantages, such as energy efficiency, quick execution, and robustness to local failures [143]. Moreover, the neuromorphic architecture employs asynchronous event-driven computing to mitigate the difficulties associated with the artificial discretization of time. This methodology is consistent with the external world’s temporal progression. Inspired by this event-driven information processing, advances in neuroscience and electronics, in both hardware and software, have made it possible to design systems that are biologically inspired. Spiking neural networks (SNNs) are often used in these systems to simulate interactive and cognitive functions [144] (a detailed overview of SNNs is provided in Section 4.3). Figure 6 provides an illustration of the differences between neuromorphic computing and traditional computing architectures.

In the discipline of neurorobotics, which includes both robotics and neuromorphic computing, bio-inspired sensors are essential for efficiently encoding sensory inputs. Furthermore, these sensors combine inputs from many sources and use event-based computation to accomplish tasks to adjust to different environmental conditions [145]. To date, however, limited study has been focused on the application of neuromorphic computing to SLAM, despite the growing availability of experimental neuromorphic processors from various companies in the last ten or so years [146]. This is primarily because practical implementations are only now beginning to become accessible.

### 4.1. Neuromorphic Computing Principles

The development of neuromorphic hardware strives to provide scalable, highly parallel, and energy-efficient computing systems [146,147]. These designs are ideal for robotic applications where rapid decision-making and low power consumption are critical since they are made to process data in real time with low latency and high accuracy [148]. Because they require vast volumes of real-time data processing, certain robotics tasks, such as visual perception and sensor fusion, are difficult for ordinary CPUs/GPUs to handle. For these kinds of activities, traditional computing architectures, such as GPUs, can be computationally and energy-intensive [28,146]. By utilizing the distributed and parallel characteristics of neural processing, neuromorphic electronics offer a solution and enable effective real-time processing of sensory data. Furthermore, conventional computing architectures do poorly on tasks requiring cognitive capacities like those of humans, such as learning, adapting, and making decisions, especially when the input space is poorly defined. In contrast, they perform exceptionally well on highly structured tasks like arithmetic computations [28].

Neuromorphic computers consist of neurons and synapses rather than a separate central processing unit (CPU) and memory units [29,149]. As their structure has gained inspiration from the working of the biological brain, the structure and function are similar to the brain where neurons and synapses are responsible for processing and memory, respectively [29]. Moreover, neuromorphic systems natively take inputs as spikes (rather than binary values) and these spikes generate the required output. The challenge to realizing the true potential of neuromorphic hardware lies with the development of a reliable computing framework that enables the programming of the complete capabilities of neurons and synapses in hardware as well as methods to communicate effectively between neurons to address the specified problems [30,31].

The advent of neuromorphic processors that employ various sets of signals to mimic the behavior of biological neurons and synapses [12,30,31] has paved a new direction in the neuroscience field. This enables the hardware to asynchronously communicate between its components and the memory in an efficient manner, which results in lower power consumption in addition to other advantages [12,29,31]. These neuromorphic systems are fully event-driven and highly parallel in contrast to traditional computing systems [29]. Today’s von Neumann CPU architectures and GPU variations adequately support artificial neural networks (ANNs), particularly when supplemented by coprocessors optimized for streaming matrix arithmetic. These conventional architectures are, however, notably inefficient in catering to the needs of SNN models [150].

### 4.2. Neuromorphic Hardware

The computation in neuromorphic systems is fundamentally based on neural networks. Neuromorphic computers are thus becoming a highly relevant platform for use in artificial intelligence and machine learning applications to enhance robustness and performance [29,32]. This has encouraged and attracted researchers [31,34,142,151,152,153] to further explore applications and development. The development of SpiNNaker (Spiking Neural Network Architecture) [143,154] and BrainScaleS [155,156] was sponsored by the European Union’s Human Brain Project to be used in the neuroscience field. Similarly, developments such as IBM’s TrueNorth [157], Intel’s Loihi [31,150], and Brainchip’s Akida [158] are some of the indications of success in neuromorphic hardware development [159].

In the following sections, recent neuromorphic developments are identified and described. Table 2 gives a summary of the currently available neuromorphic processing systems.

#### 4.2.1. SpiNNaker

The University of Manchester’s SpiNNaker project launched the first hardware platform designed specifically for SNN research in 2011 [154]. A highly parallel computer was created, SpiNNaker 2 [160], in 2018 as a part of the European Human Brain Project. Its main component is a specially made micro-circuit with 144 ARM M4 microprocessors and 18 Mbyte of SRAM. It has a limited instruction set but performs well and uses little power. Support for rate-based DNNs, specialized accelerators for numerical operations, and dynamic power management are just a few of the new features that SpiNNaker 2 offers [146].

The SpiNNaker chips are mounted on boards, with 56 chips on each board. These boards are then assembled into racks and cabinets to create the SpiNNaker neurocomputer, which has 106 processors [161]. The system functions asynchronously, providing flexibility and scalability; however, it requires the use of AER packets for spike representation through the implementation of multiple communication mechanisms.

Researchers can more successfully mimic biological brain structures with the help of SpiNNaker. It was noteworthy that it outperformed GPU-based simulations in real-time simulation for a 1 mm^2^ cortical column (containing 285,000,000 synapses and 77,000 neurons at a 0.1 ms time-step) [162]. SpiNNaker’s intrinsic asynchrony makes it easier to represent a 100 mm^2^ column by increasing the number of computing modules, a task that GPUs find challenging because of synchronization constraints.

#### 4.2.2. TrueNorth

In 2014, IBM launched the TrueNorth project, the first industrial neuromorphic device, as part of DARPA’s SyNAPSE program [163]. With 4096 neural cores that can individually simulate 256 spiking neurons in real time, this digital device has about 100 Kbits of SRAM memory for storing synaptic states. Using a digital data highway for communication, neurons encode spikes as AER packets. TrueNorth neural cores can only perform addition and subtraction; they cannot perform multiplication or division, and their functionality is fixed at the hardware level [146].

There are 256 common inputs in each neural core, which enables arbitrary connections to the 256 neurons inside the core. Because synapse weights are only encoded with two bits, learning methods cannot be implemented entirely on the chip. For running recurrent (RNN) and convolutional neural networks (CNNs) in inference mode, TrueNorth is a good choice [163]. However, to transfer learnt weights into TrueNorth configurations for learning, an extra hardware platform typically requires a GPU.

An example application from 2017 [25] uses a TrueNorth chip and DVS camera to create an event-based gesture detection system. It took 0.18 W and 0.1 s to recognize 10 gestures with 96.5% accuracy. The same researchers demonstrated an event-based stereo-vision system in 2018 [164] that boasted 200 times more energy economy than competing solutions. It used two DVS cameras and eight TrueNorth CPUs, and it could determine scene depth at 2000 disparity maps per second. Furthermore, in 2019, a scene-understanding application showed how to detect and classify several objects at a throughput of more than 100 frames per second from high-definition aerial video footage [157].

#### 4.2.3. Loihi

The first neuromorphic microprocessor with on-chip learning capabilities was introduced in 2018 with the release of Intel’s Loihi project [150]. Three Pentium processors, four communication modules, and 128 neural cores are all integrated into a single Loihi device to enable the exchange of AER packets. With 128 Kbytes of SRAM for synapse state storage, each of these cores may individually simulate up to 1024 spiking neurons. The chip can simulate up to 128,000,000 synapses and about 128,000 neurons in this setup. The mechanism smoothly maintains spike transmission from neuron to neuron and modifies its speed if the spike flow gets too strong.

Loihi allows for on-chip learning by dynamically adjusting its synaptic weights, which range from 1 to 9 bits [146]. A variable that occupies up to 8 bits and acts as an auxiliary variable in the plasticity law is included in each synapse’s state, along with a synaptic delay of up to 6 bits. Only addition and multiplication operations are required for local learning, which is achieved by weight recalculation during core configuration.

Various neurocomputers have been developed using Loihi, with Pohoiki Springs being the most potent, combining 768 Loihi chips into 24 modules to simulate 100,000,000 neurons [146]. Loihi is globally employed by numerous scientific groups for tasks like image and smell recognition, data sequence processing, PID controller realization, and graph pathfinding [31]. It is also utilized in projects focusing on robotic arm control [165] and quadcopter balancing [166].

With 128 neural cores that can simulate 120,000,000 synapses and 1,000,000 programmable neurons, Intel unveiled Loihi 2, a second version, in 2021 [31]. It integrates 3D multi-chip scaling, which enables the combining of numerous chips in a 3D environment and makes use of Intel’s 7 nm technology for a 2.3 billion transistor chip. With local broadcasts and graded spikes where spike values are coded by up to 32 bits, Loihi 2 presents a generalized event-based communication model. An innovative approach to process-based computing was presented by Intel with the launch of the Lava framework [167], an open-source platform that supports Loihi 2 implementations on CPU, GPU, and other platforms [31].

#### 4.2.4. BrainScaleS

As part of the European Human Brain Project, Heidelberg University initiated the BrainScaleS project in 2020 [168]. Its goal is to create an Application-Specific Integrated Circuit (ASIC) that can simulate spiking neurons by using analog computations. Analog computations are performed using electronic circuits, which are characterized by differential equations that mimic the activity of organic neurons. Every electronic circuit consists of a resistor and a capacitor, symbolizing a biological neuron. The second version of the 2011 release had digital processors to facilitate local learning (STDP) in addition to the analog neurons, whereas the first version did not include on-chip learning capabilities [146]. Spikes in the form of AER packets are used as a digital data highway to promote communication between neurons. A total of 130,000 synapses and 512 neurons can be simulated on a single chip.

While the analog neuron model has advantages over biological neurons (up to 10,000 times faster in analog implementation) and adaptability (compatible with classical ANNs) [169], it also has drawbacks due to its relative bulk and lack of flexibility [146]. BrainScaleS has been used to tackle tasks in a variety of fields, such as challenges involving ANNs, speech recognition utilizing SNNs, and handwritten digit recognition (MNIST) [34,170,171,172]. For example, BrainScaleS obtained a 97.2% classification accuracy with low latency, energy consumption, and total chip connections using the spiking MNIST dataset [146]. To implement on-chip learning, surrogate gradient techniques were used [173].

The use of BrainScaleS for reinforcement learning tasks using the R-STDP algorithm demonstrated the platform’s potential for local learning [174]. An Atari Ping Pong-like computer game was used to teach the system how to manipulate a slider bar [146].

#### 4.2.5. Dynamic Neuromorphic Asynchronous Processors

A group of neuromorphic systems called Dynamic Neuromorphic Asynchronous Processors (DYNAPs) were created by SynSence, a University of Zurich affiliate, using patented event-routing technology for core communication [18,146]. A significant barrier to the scalability [175] of neuromorphic systems is addressed by SynSence’s unique two-level communication model, which optimizes the ratio of broadcast messages to point-to-point communication inside neuron clusters. The research chips DYNAP-SE2 and DYNAP-SEL are part of the DYNAP family and are intended for use by neuroscientists investigating SNN topologies and communication models. Furthermore, there is DYNAP-CNN, a commercial chip designed specifically to efficiently perform SNNs that have been converted from CNNs [147]. Analog processing and digital communication are used by DYNAP-SE2 and DYNAP-SEL, whilst DYNAP-CNN is entirely digital, enabling event-based sensors (DVS) and handling image classification tasks.

DYNAP-SE2 has four cores with 65 k synapses and 1 k Leaky Integrate-and-Fire with Adaptive Threshold (LIFAT) analog spiking neurons, making it suitable for feed-forward, recurrent, and reservoir networks [146]. This chip, which offers many synapse configurations (N-methyl-D-aspartate (NMDA), α-amino-3-hydroxy-5-methyl-4-isoxazolepropionic acid (AMPA), Gamma-aminobutyric acid type A (GABAa), and Gamma-aminobutyric acid type B (GABAb)), makes research into SNN topologies and communication models easier. With five cores, including one with plastic synapses, DYNAP-SEL has a huge fan in/out network connectivity and facilitates on-chip learning. Researchers can mimic brain networks with the chip’s 1000 analog spiking neurons and up to 80,000 reconfigurable synaptic connections, of which 8000 include spike-based learning rules (STDP).

The DYNAP-CNN chip has been available with a Development Kit since 2021. It is a 12 mm^2^ chip with four million configurable parameters and over a million spiking neurons, built using 22 nm technology. It only runs in the inference mode and performs effective SNN conversion from CNNs. It achieves notable performance on applications including wake phrase identification, attentiveness detection, gesture recognition, and CIFAR-10 picture classification. There is no support for on-chip learning; the initial CNN needs to be trained on a GPU using traditional techniques like PyTorch and then converted using the Sinabs.ai framework so that it can run on DYNAP-CNN.

#### 4.2.6. Akida

Akida, created by Australian company BrainChip, stands out as the first commercially available neuromorphic processor released in August 2021 [176], with NASA and other companies participating in the early access program. Positioned as a power-efficient event-based processor for edge computing, Akida functions independently of an external CPU and consumes 100 µW to 300 mW for diverse tasks. Boasting a processing capability of 1000 frames/Watt, Akida currently supports convolutional and fully connected networks, with potential future backing for various neural network types. The chip facilitates the conversion of ANN networks into SNNs for execution.

A solitary Akida chip within a mesh network incorporates 80 Neural Processing Units, simulating 1,200,000 neurons and 10,000,000,000 synapses. Fabricated using TSMC technology, a second-generation 16 nm chip was unveiled in 2022. The Akida ecosystem encompasses a free chip emulator, the MetaTF framework for network transformation, and pre-trained models. Designing for Akida necessitates consideration of layer parameter limitations.

A notable feature of Akida is its on-chip support for incremental, one-shot, and continuous learning. BrainChip showcased applications at the AI Hardware Summit 2021, highlighting human identification after a single encounter and a smart speaker using local training for voice recognition. The proprietary homeostatic STDP algorithm supports learning, with synaptic plasticity limited to the last fully connected layer. Another demonstrated application involved the classification of fast-moving objects using an event-based approach, effectively detecting objects even when positioned off-center and appearing blurred.

### 4.3. Spiking Neural Networks

Typically, neural networks are divided into three generations, each of which mimics the multilayered structure of the human brain while displaying unique behaviors [177]. The first generation has binary (0,1) neuron output, which is derived from simple weighted synaptic input thresholding. Previous research [178] showed that networks made of artificial neurons could perform logical and mathematical operations. With the advancement of multilayer perceptron networks and the backpropagation technique, a new idea became apparent over time. In modern deep learning, this method is commonly used to overcome the shortcomings of earlier neural perceptron techniques. Artificial neural network (ANN) is the name given to this second generation. Its primary distinction from the first generation lies in neuron output, which can be a real number resulting from the weighted sum of inputs processed through a transfer function, typically sigmoidal. Weights are determined through various machine learning algorithms, ranging from basic linear regression to advanced classification.

Compared to their biological counterparts, neural networks in their first and second generations have limited modeling capabilities. Interestingly, there is no temporal reference to electrical impulses found in biological neural networks in these models. Additionally, research on biological processes remains limited. The human brain excels in processing real-time data, efficiently encoding information through various features related to spikes, including specific event times [179]. The concept of simulating neural events prompted the creation of SNNs, which currently stand as the most biologically plausible models.

An SNN architecture controls the information transfer from a presynaptic (source) neuron to a postsynaptic (target) neuron through a network of interconnected neurons connected by synapses. SNNs use spikes to encode and transport information, in contrast to traditional ANNs. Unlike a single forward propagation, each input is displayed for a predefined duration (T), resulting in several forward passes, $\frac{T}{\delta t}$. Like the biological counterpart, a presynaptic neuron transmits a signal proportionate to the synapse weight or conductance to its postsynaptic counterpart in the form of a synaptic current. Generally, when the synaptic current enters the target neuron, it causes a certain amount, δv, to change in the membrane potential ($v_{mem}$). The postsynaptic neuron fires a spike and resets its membrane voltage to the resting potential ($v_{rest}$) if the $v_{mem}$ crosses a predetermined threshold ($v_{thresh}$). On the other hand, different network topologies and applications may require different combinations of learning rules, $v_{mem}$ dynamics, and neuron models. Various methodologies can be used to describe neurons and synapse dynamics.

Compared to standard ANNs, SNNs include topologies and learning principles that closely mimic biological processes. SNNs, the third generation of ANNs, are excellent at reliable computation with little computational load. SNN neurons are not differentiable; once their states cross the firing threshold, they produce event-based spikes, but they also hold onto past states that gradually deteriorate over time. Because SNNs are dynamic, direct training with the conventional backpropagation (BP) method is difficult and considered biologically unrealistic [180]. To substitute ReLU activation functions in the ANN with the Leaky Integrate and Fire (LIF) neurons, SNNs are thus created from trained ANNs [181]. However, converted SNNs generally fail to achieve the required performance and impact the latency and power consumption. This has led to directly training SNNs using both unsupervised STDP and supervised learning rules (such as SpikeProp and gradient-based learning), which have also resulted in producing inefficient results, but the surrogate gradient learning rule was found to be effective in training complex and powerful SNNs Model [181]. Figure 7 compares the generic neuron of an artificial neural network and a spiking neural network.

Among the various neuron models proposed by the researchers, the LIF and its variants are among the most popular neuron models due to their low implementation costs [32,183]. The LIF model can be represented mathematically as follows:(3)$$C_{dt}^{dV}=-g_{L}\left(V\left(t\right)-E_{L}\right)+I(t)$$

In Equation (3), output voltage *V*(*t*) relies on the conductance $g_{L}$ of the resistor, the capacitance C of the capacitor, the resting voltage $E_{L}$, and a current source I(t). When multiplying Equation (3) by $R=\frac{1}{C}$, the $\frac{{dv}_{mem}}{d_{t}}$ in relation to the membrane time constant, $\tau_{m}$ is:(4)$$\tau_{m}\frac{{dv}_{mem}}{d_{t}}=-\left[v_{mem}\left(t\right)-v_{rest}\right]+RI$$

Consequently, the activation function $A\left(t\right)$ for LIF neurons is represented as shown in 5; $v_{mem}$ constantly decays to the rest value and undergoes a refractory period.
(5)$$A\left(t\right)=\left\{\begin{array}{c} 0,\;if\;v_{mem}<v_{thresh}\\ 1,\;if\;v_{mem}\geq v_{thresh} \end{array}\right.$$

### 4.4. Neuromorphic Computing in SLAM

Integrating neuromorphic computing into SLAM systems involves merging the distinctive characteristics of neuromorphic hardware and algorithms to enhance SLAM’s functionality. The integration needs to leverage the unique capabilities of neuromorphic computing to improve the performance and efficiency of SLAM operations [147,184,185]. This requires adapting and utilizing neuromorphic technology to address various aspects of SLAM, from sensor data processing to navigation and planning. To date, the application of neuromorphic computing in SLAM technology has had limited exploration, but its use in other related areas such as medicine, robotics and other fields has been widely studied [185,186,187,188].

In medical treatment and monitoring, neuromorphic systems are worn or implanted as components of other medical treatment tools or interface directly with biological systems [189]. This is mainly to improve diagnostic accuracy while also ensuring patient compliance. Moreover, the system can be operated to address the existing medical technology issues by offering reliable solutions that consume minimal energy, lower latency, and higher bandwidth.

The concept of brain–computer interfaces (BCIs) [190,191,192] has become popular since their initial implementations in complementary metal-oxide semiconductors (CMOSs). However, they have failed to achieve the expected efficiency. The brain–neuromorphic interface (BNI) [193] has been explored to improve and enhance BCI technology, which has contributed positively towards energy efficiency with other advantages. Similarly, in robotics, where on-board processing needs to be very compact and power-efficient, the most common existing applications of neuromorphic systems include behavior learning, locomotion control, social learning, and target learning. However, autonomous navigation tasks are the most common among neuromorphic implementations in robotics [34]. Work has also been carried out to apply biologically inspired approaches to computer vision in areas such as underwater image enhancement and visible-infrared image fusion [194,195]. Neuromorphic systems have also been used in a wide range of control applications [196,197,198,199] because these usually have strict real-time performance requirements. They are often used in real systems that have low power and small volume requirements and frequently involve temporal processing, which makes models that use recurrent connections or delays on synapses beneficial [34].

Neural network and neuromorphic implementations have been widely applied to a variety of image-based applications, such as feature extraction [200,201,202], edge detection [203,204], segmentation [205,206], compression [207,208], and filtering [209,210]. Applications such as image classification, detection, or identification are also very common [34]. Furthermore, applications of neuromorphic systems have also included the recognition of other patterns, such as pixel patterns or simple shapes [211,212]. Additionally, general character identification tasks [213,214,215] and other digit recognition tasks [216,217,218] have become highly popular. To assess the numerous neuromorphic implementations, the MNIST dataset and its variations have been employed [34,170,171,172].

Additional image classification tasks have been demonstrated on neuromorphic systems, which include the classifying of real-world images such as traffic signs [219,220,221,222], face recognition or detection [223,224,225,226,227,228], car recognition or detection [228,229,230,231,232], identifying air pollution in images [228,233,234], identifying manufacturing defects or defaults [235,236], hand gesture recognition [228,237,238], object texture analysis [239,240], and other real-world image recognition tasks [228,241]. The employment of neuromorphic systems in video-based applications has also been common [34]; video frames are analyzed as images and object recognition is performed without necessarily taking into consideration the time component [242,243,244,245]. Nevertheless, a temporal component is necessary for several additional video applications, and further works have investigated this for applications such as activity recognition [246,247,248], motion tracking [249,250], motion estimation [251,252,253], and motion detection [250].

In general, the application of neuromorphic systems has been commonly explored in the aforementioned fields as it is found to bring improvements in energy efficiency and performance as compared to traditional computing platforms [147,184,185,254]. This has led researchers to explore the incorporation of neuromorphic systems in some of the SLAM implementations, such as in [5,27,148,254,255,256], resulting in enhanced energy efficiency and performance, in addition to other benefits. In [27], when the system (which represented the robot’s 4DoF pose in a 3D environment) was integrated with a lightweight vision system (in a similar manner to the vision system of mammals), the system could generate comprehensive 3D experience maps with consistency both for simulated and real 3D environments. Using the self-learning hardware architecture (gated-memristive device) in conjunction with the spiking neurons, the SLAM system was successful in making navigation-related operations in a simple environment consuming minimal power (36 µW) [148]. Similarly, the research by [254,256] has shown that power consumption is minimal when the system employs the pose-cell array and digital head direction cell, which mimics place and head direction cells of the rodent brain, respectively. Additionally, the paper on Multi-Agent NueroSLAM (MAN-SLAM) [255], when coupled with time-domain SNN-based pose cells, was found to address the issue of [27] and also improve the accuracy of SLAM results. In a similar way, the methods reported in [5] based on ORB features combined with head direction cells and 3D grid cells was found to enhance the robustness, mapping accuracy, and efficiency of storage and operation.

Overall, the findings and results from the literature considered for this review led to two main conclusions; incorporation of neuromorphic systems or technologies in any system enhances the performance and minimizes power consumption, making it feasible for autonomous systems. However, given the resources and algorithms to fully implement the technology are still in the research stage, the full potential of the technology is yet to be realized and some researchers could only develop early system prototypes. It was determined that these technologies would require novel training and learning algorithms to be designed specifically to support them, and further work is required to develop these [34,147,188].

These previous examples demonstrate the feasibility and advantages of applying neuromorphic approaches to a range of dynamic modeling and image analysis problems, which provides strong support for the idea that integration of neuromorphic computing into SLAM systems holds significant promise for developing more capable, adaptive, and energy-efficient robotic platforms [147,184,185,254]. By leveraging the power of neuromorphic hardware and algorithms, SLAM systems should achieve enhanced performance, robustness, and scalability, paving the way for a new generation of intelligent robotic systems capable of navigating and mapping complex environments with increased efficiency and accuracy [31,185,186,187,188].

Based on this review, it is apparent that successful integration of neuromorphic processing with an event camera-based SLAM system has the potential to provide a number of benefits, including the following:Efficiency: Neuromorphic hardware is designed to mimic the brain’s parallel processing capabilities, resulting in efficient computation with low power consumption. This efficiency is particularly beneficial in real-time SLAM applications where rapid low-power processing of sensor data is crucial.Adaptability: Neuromorphic systems can adapt to and learn from their environment, making them well-suited for SLAM tasks in dynamic or changing environments. They can continuously update their internal models based on new sensory information, leading to improved accuracy and robustness over time.Event-Based Processing: Event cameras capture data asynchronously in response to changes in the environment. This event-based processing enables SLAM systems to focus computational resources on relevant information, leading to faster and more efficient processing compared to traditional frame-based approaches.Sparse Representation: Neuromorphic algorithms can generate sparse representations of the environment, reducing memory and computational requirements. This is advantageous in resource-constrained SLAM applications, such as those deployed on embedded or mobile devices.

While neuromorphic computing holds promise for enhancing SLAM capabilities [147,184,185], several challenges will need to be overcome to fully exploit its potential in real-world applications [34,147,188]. Collaboration between researchers in neuromorphic computing, robotics, and computer vision will be crucial in addressing these challenges and realizing the benefits of neuromorphic SLAM systems. One challenge is that neuromorphic hardware is still in the early stages of development, and integration of neuromorphic computing into SLAM systems may require custom hardware development or significant software adaptations. More significantly, adapting existing SLAM algorithms for implementation on neuromorphic hardware is a complex task that requires high levels of expertise in robotics and neuromorphic systems [34,147,188]. Significant research effort will be required to develop and refine these neuromorphic algorithms before outcomes having a comparable level to those of current state-of-the-art SLAM systems can be achieved.

## 5. Conclusions

SLAM based on event cameras and neuromorphic computing represents an innovative approach to spatial perception and mapping in dynamic environments. Event cameras capture visual information asynchronously, responding immediately to changes in the scene with high temporal resolution. Neuromorphic computing, inspired by the brain’s processing principles, has the capacity to efficiently handle these event-based data, enabling real-time, low-power computation. By combining event cameras and neuromorphic processing, SLAM systems could achieve several advantages, including low latency, low power consumption, robustness to changing lighting conditions, and adaptability to dynamic environments. This integrated approach offers efficient, scalable, and robust solutions for applications such as robotics, augmented reality, and autonomous vehicles, with the potential to transform spatial perception and navigation capabilities in various domains.

### 5.1. Summary of Key Findings

VSLAM systems based on traditional image sensors such as monocular, stereo, or RGB-D cameras have gained significant development attention in recent decades. These sensors can gather detailed data about the scene and are available at affordable prices. They also have relatively low power requirements, making them feasible for autonomous systems such as self-driving cars, unmanned aerial vehicles, and other mobile robots. VSLAM systems employing these sensors have achieved reasonable performance and accuracy but have often struggled in real-world contexts due to high computational demands, limited adaptability to dynamic environments, and susceptibility to motion blur and lighting changes. Moreover, they face difficulties in real-time processing, especially in resource-constrained settings like autonomous drones or mobile robots.

To overcome the drawbacks of these conventional sensors, event cameras have begun to be explored. They have been inspired by the working of biological retinas and attempt to mimic the characteristics of human eyes. This biological design influence for event cameras means they consume minimal power and operate with lower bandwidth, in addition to other notable features such as very low latency, high temporal resolution, and wide dynamic range. These attractive features make event cameras highly suitable for robotics, autonomous vehicles, drone navigation, and high-speed tracking applications. However, they operate on a fundamentally different principle compared to traditional cameras; event cameras respond to the brightness changes of the scene and generate events rather than capturing the full frame at a time. This poses challenges as algorithms and approaches employed in conventional image processing and SLAM systems cannot be directly applied and novel methods are required to realize their potential.

For SLAM systems based on event cameras, relevant methods can be selected based on the event representations and the hardware platform being used. Commonly employed methods for event-based SLAM are featured-based, direct, motion-compensated, and deep learning approaches. Feature-based methods can be computationally efficient as they only deal with the small numbers of events produced by the fast-moving cameras for processing. However, their efficacy diminishes while dealing with a texture-less environment. On the other hand, the direct method can achieve robustness in a texture-less environment, but it can only be employed for moderate camera motions. Motion-compensated methods can offer robustness in high-speed motion as well as in large-scale settings, but they can only be employed for rotational camera motions. Deep learning methods can be effectively used to acquire the required attributes of the event data and generate the map while being robust to noise and outliers. However, this requires large amounts of training data, and performance cannot be guaranteed for different environment settings. SNNs have emerged in recent years as alternatives to CNNs and are considered well-suited for data generated by event cameras. The development of practical SNN-based systems is, however, still in the early stages and relevant methods and techniques need considerable further development before they can be implemented in an event camera-based SLAM system.

For conventional SLAM systems, traditional computing platforms usually require additional hardware such as GPU co-processors to perform the heavy computational loads, particularly when deep learning methods are employed. This high computational requirement means power requirements are also high, making them impractical for deployment in mobile autonomous systems. However, neuromorphic event-driven processors utilizing SNNs to model cognitive and interaction capabilities show promise in providing a solution. The research on implementing and integrating these emerging technologies is still in the early stages; however, an additional research effort will be required to realize this potential.

This review has identified that a system based on event cameras and neuromorphic processing presents a promising pathway for enhancing state-of-the-art solutions in SLAM. The unique features of event cameras, such as adaptability to changing lighting conditions, support for high dynamic range, and lower power consumption due to the asynchronous nature of event data generation, are the driving factors that can help to enhance the performance of the SLAM system. In addition, neuromorphic processors, which are designed to efficiently process and support parallel incoming event streams, can help to minimize the computational cost and increase the efficiency of the system. Such a neuromorphic SLAM system has the possibility of overcoming significant obstacles in autonomous navigation, such as the need for quick and precise perception, while simultaneously reducing problems relating to real-time processing requirements and energy usage. Moreover, if appropriate algorithms and methods can be developed, this technology has the potential to transform the realm of mobile autonomous systems by enhancing their agility, energy efficiency, and ability to function in a variety of complex and unpredictable situations.

### 5.2. Limitations of the Study

This study concluded that the integration of event cameras and neuromorphic computing approaches in SLAM technology has the potential to enhance performance and robustness. However, it should be acknowledged that this study has been conducted based on the existing literature, which is mostly theoretical and simulation-based because of the limited availability of required resources (e.g., neuromorphic hardware). Moreover, stable and suitable algorithms (e.g., SNN models) for these emerging technologies are yet to be fully developed and the results from the existing models could be biased. Additionally, the study is solely based on qualitative analysis of the related work and practical demonstrations have yet to be performed, although efforts are underway to remedy this in the future.

### 5.3. Current State-of-the-Art and Future Scope

During the last few decades, much research has focused on implementing SLAM systems based on frame-based cameras and laser scanners. Nonetheless, a fully reliable and adaptable solution has yet to be discovered due to the computational complexities and sensor limitations, leading to systems requiring high power consumption and having difficulty adapting to changes in the environment, rendering them impractical for many use cases, particularly for mobile autonomous systems. For this reason, researchers have begun to shift focus to finding alternative or new solutions to address these problems. One promising direction for further exploration was found to be the combination of an event camera and neuromorphic computing technology due to the unique benefits that these complementary approaches can bring to the SLAM problem.

The research to incorporate event cameras and neuromorphic computing technology into a functional SLAM system is, however, currently in the early stages. Given that the algorithms and techniques employed in conventional SLAM approaches are not directly applicable to these emerging technologies, the necessity of finding new algorithms and methods within the neuromorphic computing paradigm is the main challenge faced by researchers. Some promising approaches to applying event cameras to the SLAM problem have been identified in this paper, but future research focus needs to be applied to the problem of utilizing emerging neuromorphic processing capabilities to implement these methods practically and efficiently.

### 5.4. Neuromorphic SLAM Challenges

Developing SLAM algorithms that effectively utilize event-based data from event cameras and harness the computational capabilities of neuromorphic processors presents a significant challenge. These algorithms must be either heavily modified or newly conceived to fully exploit the strengths of both technologies. Furthermore, integrating data from event cameras with neuromorphic processors and other sensor modalities, such as IMUs or traditional cameras, necessitates the development of new fusion techniques. Managing the diverse data formats, temporal characteristics, and noise profiles from these sensors, while maintaining consistency and accuracy throughout the SLAM process, will be a complex task.

In terms of scalability, expanding event cameras and neuromorphic processor-based SLAM systems to accommodate large-scale environments with intricate dynamics will pose challenges in computational resource allocation. It is essential to ensure scalability while preserving real-time performance for practical deployment. Additionally, event cameras and neuromorphic processors must adapt to dynamic environments where scene changes occur rapidly. Developing algorithms capable of swiftly updating SLAM estimates based on incoming event data while maintaining robustness and accuracy is critical.

Leveraging the learning capabilities of neuromorphic processors for SLAM tasks, such as map building and localization, necessitates the design of training algorithms and methodologies proficient in learning from event data streams. The development of adaptive learning algorithms capable of enhancing SLAM performance over time in real-world environments presents a significant challenge. Moreover, ensuring the correctness and reliability of event camera and neuromorphic processor-based SLAM systems poses hurdles in verification and validation. Rigorous testing methodologies must also be developed to validate the performance and robustness of these systems. If these challenges can be overcome, the potential rewards are significant, however.

## Figures and Tables

**Figure 1 biomimetics-09-00444-f001:**
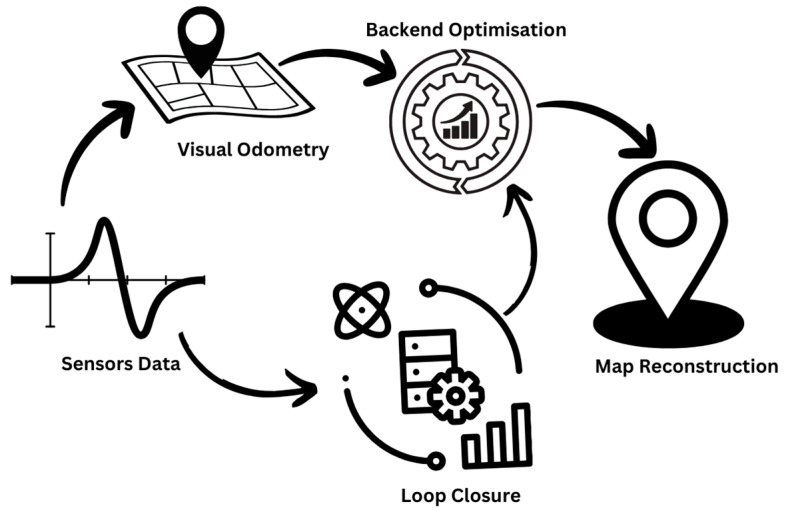
Classic visual SLAM framework.

**Figure 2 biomimetics-09-00444-f002:**
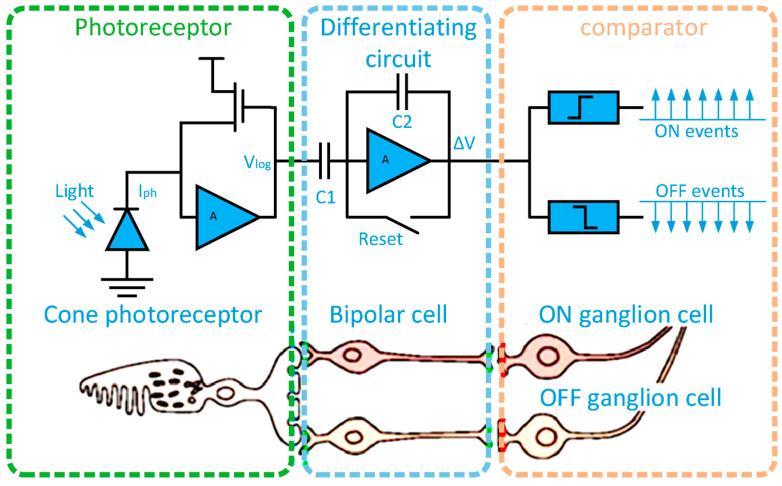
Three-layer model of a human retina and corresponding event camera pixel circuitry (adapted from [26]).

**Figure 3 biomimetics-09-00444-f003:**
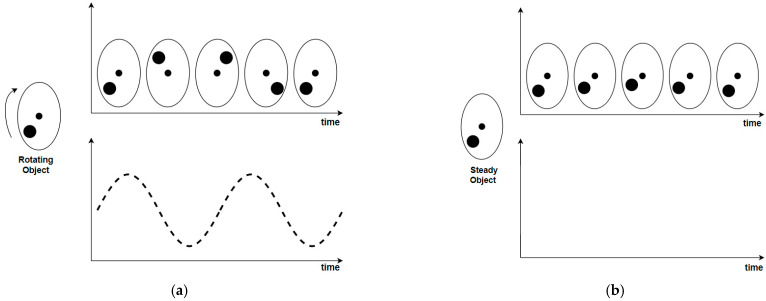
The output of a traditional frame-based camera (upper) vs. an event camera (lower): (**a**) with a rotating object; (**b**) with a steady object (constant brightness); in this case, no events are generated as depicted by the empty final graph [18].

**Figure 4 biomimetics-09-00444-f004:**
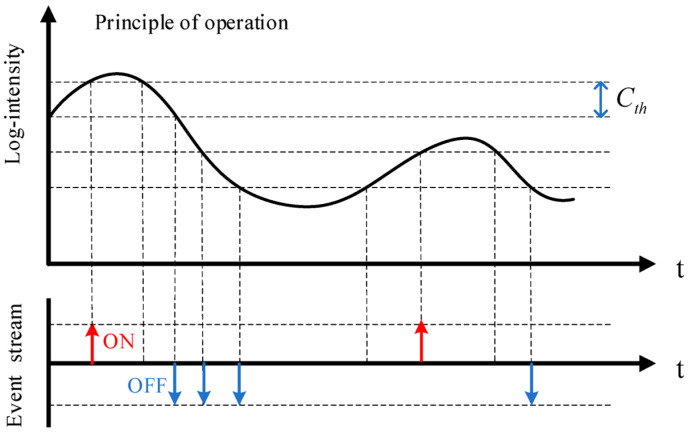
The process of generating events by the event camera (adapted from [26]); the ON (upward) arrow represents an increase and the OFF (downward) arrow depicts a decrease in log intensity value than the threshold.

**Figure 5 biomimetics-09-00444-f005:**
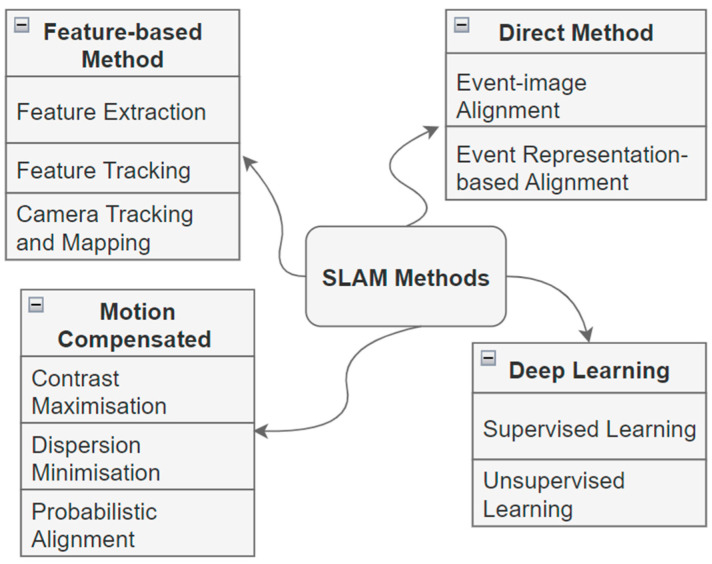
Common methods used in event-based SLAM.

**Figure 6 biomimetics-09-00444-f006:**
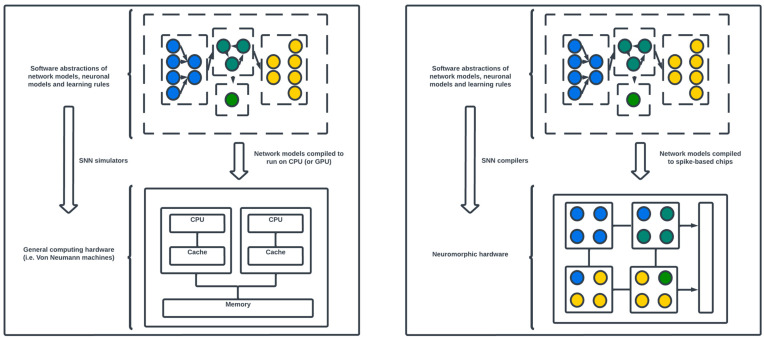
Neuromorphic computing vs. von Neuman computing architecture (adapted from [35]).

**Figure 7 biomimetics-09-00444-f007:**
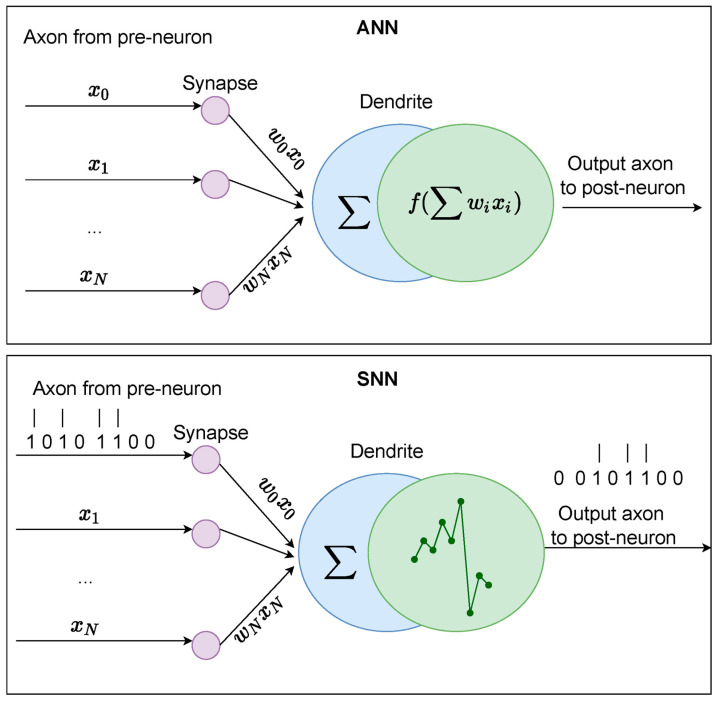
A comparison between neurons of ANNs and SNNs (adapted from [182]).

**Table 1 biomimetics-09-00444-t001:** Classification of VSLAM algorithms/methods.

Year	Name	Sensors	Descriptions (Key Points)	Strength (Achievements)
2024	TextSLAM [47]	RGB-D	Text objects in the environment are used to extract semantic features	More accurate and robust even under challenging conditions
2023	HFNet-SLAM [8]	Monocular	Extension of ORB-SLAM3 (incorporates CNNs)	Performs better than ORB-SLAM3 (higher accuracy)
2022	SO-SLAM [48]	Monocular	Introduced object spatial constraints (object level map)	Proposed two new methods for object SLAM
2022	SDF-SLAM [49]	Monocular	Semantic deep fusion model with deep learning	Less absolute error than the state-of-the-art SLAM framework
2022	UV-SLAM [50]	Monocular	Vanishing points (line features) are used for structural mapping	Localization accuracy and mapping quality have improved
2021	RS-SLAM [51]	RGB-D	Employed semantic segmentation model	Both static and dynamic objects are detected
2021	RDMO-SLAM [52]	RGB-D	Semantic label prediction using dense optical flow	Reduce the influence of dynamic objects in tracking
2021	RDS-SLAM [53]	RGB-D	Extends ORB-SLAM3; Added semantic thread and a semantic-based optimization thread	Tracking thread is not required to wait for semantic information as novel threads run in parallel
2021	ORB-SLAM3 [54]	Monocular, Stereo and RGB-D	Perform visual, visual-inertial and multimap SLAM	Effectively exploits the data associations and boosts the system accuracy level
2020	Structure-SLAM [55]	Monocular	Decoupled rotation and translation estimation	Outperforms the state of the art on common SLAM benchmarks
2020	VPS-SLAM [56]	RGB-D	Combined low-level VO/VIO with planar surfaces	Provides better results than the state-of-the-art VO/VIO algorithms
2020	DDL-SLAM [46]	RGB-D	Dynamic object segmentation and background painting added to ORB-SLAM2	Dynamic objects detected utilizing semantic segmentation and multi-view geometry
2019	PL-SLAM [57]	Stereo	Combines point and line segments	The first open-source SLAM system with points and line segment features
2017	ORB-SLAM2 [58]	Monocular, Stereo and RGB-D	Complete SLAM system including map reuse, loop closing, and re-localization capabilities	Achieved state-of-the-art accuracy while evaluating 29 popular public sequences
2015	ORB-SLAM [59]	Monocular	Feature-based monocular SLAM system	Robust to motion clutter, allows wide baseline loop closing and re-localization
2014	LSD-SLAM [60]	Monocular	Direct monocular SLAM system	Achieved post-estimation accuracy and 3D environment reconstructions
2011	DTAM [44]	Monocular	Camera tracking and reconstruction based on a dense feature	Achieved real-time performance using the commodity GPU hardware
2007	PTAM [43]	Monocular	Estimate camera pose in an unknown scene	Accuracy and robustness have surpassed the state-of-the-art system
2007	MonoSLAM [42]	Monocular	Real-Time Single Camera SLAM	Recovered the 3D trajectory of a monocular camera

**Table 2 biomimetics-09-00444-t002:** Comparison of existing neuromorphic processor architectures.

Year	Processor/ Chips	I/O	On-Device Training	Feature Size (nm)	Remarks
2011	SpiNNaker	Real Numbers, Spikes	STDP	22	Minimal power consumption (20 nj/operation). First successful mimicking of biological brain-like structure. AER packets are required to be used for spike representations.
2014	TrueNorth	Spikes	No	28	First industrial neuromorphic device. Functionality is fixed at the hardware level; only addition and subtraction can be performed.
2018	Loihi	Spikes	STDP	14	First neuromorphic processor with on-chip learning capabilities. Spike signals are not programmable and lack context or range of values.
2020	BrainScaleS	Real Numbers, Spikes	STDP, Surrogate Gradient	65	Simulates spiking neurons using analog circuitry. Perform faster than biological neurons but lack flexibility.
2021	Loihi2	Real Numbers, Spikes	STDP, Surrogate, Backpropagation	7	Integrates 3D multi-chip scaling that enables it to combine with numerous chips. Lava software framework was launched to streamline Loihi2 implementations. Limits the size and complexity of neural networks due to resource constraints.
2021	DYNAP SE2, SEL, CNN	Spikes	STDP (SEL)	22	DYNAP-SE2 is suitable for feed-forward, recurrent and reservoir networks. DYNAP-SEL facilitates on-chip learning. DYNAP-CNN supports conversion of CNN to SNN.
2021	Akida	Spikes	STDP (Last Layer)	28	First commercially available neuromorphic chip. Facilitates conversion of CNN to SNN. Notable features are on-chip, one-shot and continuous learning. Only the last layer of the fully connected layer supports on-chip continual learning.

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
