# Peer review of "Application of Event Cameras and Neuromorphic Computing to VSLAM: A Survey"

_biomimetics, 2024, doi:10.3390/biomimetics9070444_

Round 1

Reviewer 1 Report

Comments and Suggestions for Authors

This paper surveys the application of event cameras and neuromorphic computing to VSLAM, which is timely and helpful. The article still has the following issues that need further improvement.

 1.    The paper extensively cites another review article [19], and the review work should be independently summarized;

 2.  A large section of text lacks references and only describes "Several techniques...", which is inappropriate. Corresponding references should be provided, such as lines 525-551 and 759-811.

3.    Although the paper cites a large number of references, it is still incomplete, such as the following references not being cited:

Huang, Tiejun, et al. "1000× faster camera and machine vision with ordinary devices." Engineering 25 (2023): 110-119.

4. The paper lacks charts, and the working methods of the cited discourse should be represented through graphics and compared using charts. To improve the readability of the paper.

5. Sections 4.1 and 4.2 are too long.

Comments on the Quality of English Language

The language description needs further improvement.

Author Response

Thank you very much for your kind feedback and suggestions that have helped to improve our work. 

Responses to the comments have been attached for your kind reference. 

Reviewer 2 Report

Comments and Suggestions for Authors

The paper offers a comprehensive, albeit general overview of the merits and applications potential of integrating event cameras and neuromorphic processors into VSLAM systems. Recent research findings are reported through an extensive literature survey, with the presentation being coherent and well structured. Moreover, the paper is excellently written and very readable. I recommend publication of the paper in its present form.

Author Response

We thank you for this positive endorsement.

Reviewer 3 Report

Comments and Suggestions for Authors

The submitted review paper presents a comparative study of approaches for applying event cameras and neuroinspiring computing to visual simultaneous localization and mapping. It is an interesting subject for the readers of the Biomimetics journal. The submitted manuscript is well-written and easy to understand.

But this actual version needs to be improved. This reviewer has the following considerations:

1. The introduction lacks background information on VSLM and needs to highlight the importance of this review.

2. In review papers, it is necessary to describe the methods used to search for and select the relevant literature, also is necessary to include the procedures to find manuscript using editorial databases, keywords, and criteria for inclusion/exclusion of the papers.

3. It is also important to present a logical flow of ideas to guide the reader through the review.

4. After that, it is important to summarize each study's main findings and identify patterns, trends, or inconsistencies across the literature. The use of figures can support the readers to understand the review discussion.

5. Another important aspect is the discussion of future trends and areas where further research is needed. Identifying limitations in existing studies is another important point in the review paper.

6. the authors critically reflected on the reviewed literature's strengths and weaknesses. Still, it is interesting to make a discussion of any biases or limitations in the existing research are also important.

Author Response

(The authors gave the same response as above.)

Round 2

Reviewer 3 Report

Comments and Suggestions for Authors

The authors addressed all the reviewer's concerns. I have no further comment. 

Author Response

Responses are all given in the attached file. 
